# Sex Differences in Conversion Risk from Mild Cognitive Impairment to Alzheimer’s Disease: An Explainable Machine Learning Study with Random Survival Forests and SHAP

**DOI:** 10.3390/brainsci14030201

**Published:** 2024-02-22

**Authors:** Alessia Sarica, Assunta Pelagi, Federica Aracri, Fulvia Arcuri, Aldo Quattrone, Andrea Quattrone

**Affiliations:** Neuroscience Research Center, Department of Medical and Surgical Sciences, Magna Graecia University, 88100 Catanzaro, Italy; assunta.pelagi@studenti.unicz.it (A.P.);

**Keywords:** Alzheimer’s disease, random survival forests, sex differences

## Abstract

Alzheimer’s disease (AD) exhibits sex-linked variations, with women having a higher prevalence, and little is known about the sexual dimorphism in progressing from Mild Cognitive Impairment (MCI) to AD. The main aim of our study was to shed light on the sex-specific conversion-to-AD risk factors using Random Survival Forests (RSF), a Machine Learning survival approach, and Shapley Additive Explanations (SHAP) on dementia biomarkers in stable (sMCI) and progressive (pMCI) patients. With this purpose, we built two separate models for male (M-RSF) and female (F-RSF) cohorts to assess whether global explanations differ between the sexes. Similarly, SHAP local explanations were obtained to investigate changes across sexes in feature contributions to individual risk predictions. The M-RSF achieved higher performance on the test set (0.87) than the F-RSF (0.79), and global explanations of male and female models had limited similarity (<71.1%). Common influential variables across the sexes included brain glucose metabolism and CSF biomarkers. Conversely, the M-RSF had a notable contribution from hippocampus, which had a lower impact on the F-RSF, while verbal memory and executive function were key contributors only in F-RSF. Our findings confirmed that females had a higher risk of progressing to dementia; moreover, we highlighted distinct sex-driven patterns of variable importance, uncovering different feature contribution risks across sexes that decrease/increase the conversion-to-AD risk.

## 1. Introduction

Alzheimer’s disease (AD) is a neurodegenerative pathology that differentially affects women and men [1,2,3,4,5], where women have a higher prevalence than men, representing two-thirds of AD patients in the US [3]. Many hypotheses exist about sex differences in the progression from Mild Cognitive Impairment (MCI) to AD, but the literature reports heterogeneous findings [2,6]. Generally, the higher prevalence of AD in women has been associated with longer female life expectancy and with biases in patient enrollment [7]. However, other studies showed a more complex picture, focusing on the neurobiological vulnerability of women, probably related to sex hormones, like estrogen [5,7]. Regarding the psychosocial aspects, women are more prone to life stress, social isolation, and insomnia [8], and their vulnerability to stressful events is enhanced by genes like the APOE e4 allele [9]. In a non-cognitively impaired population, women demonstrate higher scores in verbal tasks and slower cognitive decline than men at all ages, while men perform better than women in visuospatial and motor coordination tasks [2]. Differences in verbal memory tasks are lost at the early stage of AD [2], although other works found that these differences also persist at early stages [10,11]. However, the literature agrees in affirming that women lose their better verbal memory performance when dementia is diagnosed [2,10,11]. A recent study [4] that performed a sex-stratified analysis found that auditory verbal memory and difficulties in activities of daily living are stronger risk factors for women than men in predicting the progression from MCI to AD. Regarding neuroimaging evidence, women showed a faster rate of brain atrophy than men [2,4], and in particular, hippocampal volume changes in women compared to men had a more prominent contribution to the progression from a normal cognitive state to MCI or AD [5]. The rate of changes in white matter hyperintensities also showed sex-linked characteristics, affecting more men than women when progressing to AD [5]. Two other recent studies demonstrated the vulnerability of women to AD pathology: the first one [12] explored brain glucose metabolism and the plasma beta-amyloid 42/40 ratio, and the second one [13] investigated a combination of functional and structural markers.

On the other hand, several works showed that no sex differences in the progression to AD exist and that the risk is equal between males and females [6,14], contradicting the hypothesis of sexual dimorphism in dementia [15], and stating that those differences are due only to the longer life expectancy of women.

Given these discrepancies in the literature, we aimed to investigate the sex differences in the risk prediction of conversion from MCI to AD using a Machine Learning survival approach, Random Survival Forests (RSF) [16,17]. RSF is an adaptation of the Random Forests (RF) [18,19] algorithm to handle right-censored data and to provide the assessment of survival probability and risk, which is fully nonparametric and thus independent from data distribution; it can handle multicollinearity and it intrinsically provides feature selection [16,17]. RSF showed stability and robustness when trained on multi-modal data, and it had better performance on biomedical datasets [20,21] as well as on dementia data [22,23] compared to statistical approaches like the Cox Proportional Hazard [24] (CPH). Moreover, we demonstrated in [23] that the black-box nature of RSF and its poor explainability could be overcome through the model-agnostic method Shapley Additive Explanations (SHAP) [25]. SHAP provides a unified framework to interpret ML predictions based on game theory, which assigns to each feature a Shapley value that represents its average marginal contribution to the predicted risk across all possible feature coalitions [23,26].

In the present work, we used a dataset from the Alzheimer’s Disease Neuroimaging Initiative (ADNI) consisting of well-known dementia biomarkers [22,23], such as clinical, cognitive, cerebrospinal fluid (CSF) and imaging features, of stable MCI patients (sMCI) and progressive MCI patients (pMCI), who change their diagnosis to AD over time. In detail, we applied RSF and SHAP separately on the male and female MCI cohorts to predict the risk of conversion to AD, and more importantly to assess whether global and local explanations differ between the sexes. Differences in the explanations of male and female models were quantified using the Rank-Biased Overlap [27] (RBO), which has been used in survival analysis [22,23] to estimate the overlap between ML feature importance by varying the number of the top variables that are considered as important [28]. Finally, we investigated the individual predictions of male and female MCI patients, stratified by high-, medium-, and low-risk grades, using SHAP waterfall plots, which provide a highly intelligible overview of the variable contribution to the decrease or increase in the conversion-to-AD risk.

## 2. Materials and Methods

### 2.1. Dataset Preparation

Data used in the preparation of this article were obtained from the Alzheimer’s Disease Neuroimaging Initiative (ADNI) database (adni.loni.usc.edu). The ADNI was launched in 2003 as a public–private partnership, led by Principal Investigator Michael W. Weiner, MD. The primary goal of ADNI has been to test whether serial magnetic resonance imaging (MRI), positron emission tomography (PET), other biological markers, and clinical and neuropsychological assessment can be combined to measure the progression of Mild Cognitive Impairment (MCI) and early Alzheimer’s disease (AD).

ADNI enrolls participants between the ages of 55 and 90 who are recruited at 57 sites in the United States and Canada. After obtaining informed consent, participants undergo a series of initial tests that are repeated at intervals over subsequent years, including a clinical evaluation, neuropsychological tests, genetic testing, lumbar puncture, and MRI and PET scans. Details about the inclusion/exclusion criteria and about the enrollment procedure can be found on the ADNI website.

Data table files (*csv*) from ADNI were downloaded on 5 June 2023, and they were as follows: DXSUM_PDXCONV_ADNIALL, ADNIMERGE, NEUROBAT, CDR, GDSCALE, FAQ, MMSE, ADASSCORES, UPENNBIOMK_MASTER_FINAL (9, 10, 12), and BAIPETNMRC_04_12_18. The software KNIME 4.6.1 [29] was used to filter and join these tables. Details about the dataset preparation can be found elsewhere [22,23]. Briefly, the final dataset used for the ML analysis included patients whose diagnosis changed over time from MCI to AD (pMCI) and patients who maintained their baseline diagnosis as stable MCI (sMCI). The *event* or *censorship* occurrence was a binary variable, where 1 (pMCI patient) represents the event of conversion from MCI to AD, and 0 represents censorship (sMCI patient). The *time* variable represented the number in months (m06, m12, m18, m24, m36, and m48) after the baseline visit in which the event/censorship occurred. The time interval ranged from 6 months to 36 months (3 years), which was different from [23] because data for month 48 were unusable due to the low sample size (16 males; 6 females). All data and all subjects were from the ADNI1 protocol and collected at baseline or the screening visit. The description of demographics, clinical, neuropsychological, and neuroimaging features are reported in Appendix A.

The final dataset consisted of 365 subjects that were split by sex: 233 males divided into 136 sMCI (M-sMCI) and 97 pMCI (M-pMCI), and 132 females divided into 62 sMCI (F-sMCI) and 70 pMCI (F-pMCI).

Categorical variables (PTETHCAT, PTRACCAT, PTMARRY) were converted to numerical data using the One-Hot Encoding approach [30,31] (python function *get_dummies()*). Missing data were imputed using the missForest algorithm [32] (python package *missingpy* 0.2.0), which demonstrated less error than statistical imputation methods on dementia [33] and Parkinson’s disease [34] data. The descriptive statistics of the dataset stratified by sex are reported in Table 1.

### 2.2. Statistical Analysis

Statistical analyses were performed to compare features between male sMCI and male pMCI patients (M-sMCI vs. M-pMCI), and female sMCI and female pMCI patients (F-sMCI vs. F-pMCI). Moreover, we compared male sMCI with female sMCI patients (M-sMCI vs. F-sMCI), and male pMCI with female pMCI patients (M-pMCI vs. F-sMCI).

Analysis of variance (ANOVA) was employed to assess differences between groups in terms of age and years of education, while the Chi-square test was applied to evaluate differences in the distributions of categorical variables. Analysis of covariance (ANCOVA) with age as a covariate was employed for clinical and cognitive variables, while for neuroimaging features, ANCOVA had age and ICV (significant at *p* < 0.05). All statistical tests were implemented with Python 3.8 and the package *scikit-learn* 1.1.3.

### 2.3. Random Survival Forests

Various studies have assessed the efficacy of ML techniques for dementia survival analysis, especially for predicting the conversion risk from MCI to AD [31,35,36,37,38,39]. Most of them showed that Random Survival Forests [17] had better performance than the classical statistical approaches like Cox Proportional Hazard, or other methods based on Random Forests [18,19]. In particular, we demonstrated in [22] that RSF had higher accuracy than Conditional Survival Forest (CSF) [40] and Extra Survival Trees (XST) [41] in predicting the conversion-to-AD risk on dementia biomarkers from ADNI. Moreover, we showed in [23] that the clinical utility of RSF can be boosted through SHAP to enhance its interpretability. The strengths of RSF rely on the robustness to outliers, no convergence issues, preservation from overfitting thanks to out-of-bag (cross-validated) prediction, the reliable inference of training data, and particularly its intrinsic variable importance measure, which is fully nonparametric and independent from data distributions [17].

In detail, RSF follows the same principles of RF [18,19] for growing decision trees, and when splitting tree nodes, it applies bootstrapping and random feature selection. The rule for splitting a node is based on the log-rank test statistic to maximize the survival difference between daughter nodes. For each node in the tree, the null hypothesis that there is no difference between the two groups in the probability of an event is tested. The ensemble’s cumulative hazard is estimated with cumulative hazard functions calculated for each tree, while out-of-bag (OOB) estimators are used to assess the prediction accuracy and the variable importance [16,17,22].

### 2.4. Machine Learning Analysis

A forked repository (https://github.com/bacalfa/pysurvival/, Bacalfa) from the python package *PySurvival* (https://square.github.io/pysurvival/, Fotso et al., 2019) was used to conduct survival analyses to have the compatibility of the RSF algorithm implementation with the sklearn package (accessed on 1 December 2023). The package *seaborn* (0.12.2) was employed to modify the original plotting functions of PySurvival.

Two RSF models were built separately, one trained only on male MCI patients (M-RSF) and the second trained only on female MCI patients (F-RSF), with the same procedure in [23] and as described below. Datasets were randomly split with a static seed into training and test sets (80–20%) stratified by the column *event* and *time* to maintain the original distribution of occurrences, obtaining 109 sMCI and 77 pMCI in the training set and 27 sMCI and 20 pMCI in the test set for the male group, and 50 sMCI and 55 pMCI in the training set and 12 sMCI and 15 pMCI in the test set for the female group. Hyperparameter tuning was applied to maximize the performance on the training set through a randomized search (RandomizedSearchCV) with a 3-fold cross-validation (*cv*) and 50 repetitions [20,22,23,31]. RSF hyperparameters were as follows: maximum depth (max_depth), minimum number of samples required to be at a leaf node (min_node_size), number of features to consider when looking for the best split (max_features), and percentage of original samples used in each tree building (sample_size_pct). As described in [22,23], the number of trees was kept static at 200, and importance mode (importance_mode) was set to permutation for both M-RSF and F-RSF to allow their comparison.

The performance of RSF models was evaluated using Harrell’s concordance index (*c*-index) [42] on training sets (with a 5-fold cross-validation) and on test sets. The *c*-index was born to generalize the area under the ROC curve (AUC) in the presence of right-censored data and for the survival analysis; the model has an almost perfect discriminatory power if its value is close to 1, while it has no ability to discriminate between low- and high-risk subjects if it is close to 0.5 (random prediction) [22,23]. In addition to the c-index, we evaluated the accuracy of the predicted survival function on the test set across multiple timepoints with the Integrated Brier score (IBS) [43]. The IBS value is between 0 and 1, where 0 is for a perfect model, while a cut-off limit of 0.25 is considered as critical [43].

The estimated survival time curve of test sets was obtained using the Kaplan–Meier method (KM) [44] and visually compared with predicted survival curves determined by the M-RSF and F-RSF models. Deviations from KM curves were quantified using the Root Mean Square Error (RMSE) and median/mean absolute error.

#### 2.4.1. Global Explanation

In addition to the feature importance provided intrinsically by the two models M-RSF and F-RSF, we evaluated the permutation importance [23], which is defined as an increase in the prediction error when a feature’s value is randomly shuffled. Permutation importance was implemented using ELI5 [45] with 50 repetitions (python package scikit-learn 1.3.0) [23].

As a further global explanation, we employed Shapley Additive Explanation [25] (SHAP, python package *SHAP* 0.42.1), which is a model-agnostic unified framework based on game theory to interpret ML classification predictions and has also been recently applied for survival analysis [23]. Two SHAP explainers (*shap.Explainer*) were fit separately on predicted risk scores of training sets by M-RSF and by F-RSF (function *predict_risk* by pysurvival).

A pairwise similarity between the global explanations of the two models, M-RSF and F-RSF, was quantitively evaluated using the Rank-Biased Overlap [27] (RBO, python package *rbo* v.0.1.2, https://github.com/changyaochen/rbo accessed on 1 December 2023), which can assume values in the range [0, 1], where 0 means disjoint and 1 means identical. RBO has been used in survival analysis [22,23] to estimate the overlap between ML feature rankings by varying the number of top variables considered as important (depths *d*) [28].

No feature selection was applied since the recent literature on survival analysis showed no improvement in performance [20,23,26]. In the same way, we kept correlated variables, since it has been demonstrated that multicollinearity did not perturb SHAP explanations of RSF [23].

#### 2.4.2. Local Explanation

Local explanations of both M-RSF and F-RSF models were explored with SHAP on test sets. Individual predictions determined by M-RSF and F-RSF were used to manually stratify male and female pMCI test patients according to their conversion-to-AD risk score (low, medium, and high) [22]. Then, we estimated the cumulative density function of six randomly selected pMCI patients, one male and one female per risk grade (M-pMCI#1 and F-pMCI#1 high risk, M-pMCI#2 and F-pMCI#2 medium risk, M-pMCI#3 and F-pMCI#3 low risk), and one stable MCI test subject per sex (M-sMCI and F-sMCI with a numeric risk score lower than 1). These test subjects were finally studied with SHAP waterfall plots.

## 3. Results

The results of statistical analysis between male and female MCI patients are reported in Table 2. Regarding the analysis of the male group, sMCI and pMCI patients had significantly different values in almost all features, except for age, education level, RAVLT forgetting, mPACCdigit, mPACCtrailsB, GDTOTAL, BNTTOTAL, TAU, PTAU, Ventricles, WholeBrain, and ICV (*p* > 0.05). In the female group, sMCI and pMCI patients had a higher number of statistically insignificant comparisons than the male group. Features without differences between female sMCI and female pMCI patients were age, education level, CDRSB, RAVLT forgetting, DIGITSCOR, TRABSCOR, mPACCdigit, mPACCtrailsB, GDTOTAL, COPYSCOR, BNTTOTAL, TAU, PTAU, Ventricles, WholeBrain, and ICV (*p* > 0.05). In the comparison between male and female sMCI patients, only education level and ABETA42 were significantly different, while no other features showed differences in the comparison between male and female pMCI patients (Table 2).

Table 3 reports the results of hyperparameter tuning obtained through a randomized search. Optimal hyperparameter values provided a *c*-index (mean of 3-fold *cv* with 50 repetitions) of 0.839 for M-RSF and 0.804 for the F-RSF. Regarding the performance of best models, M-RSF reached high values of the *c*-index both on the test set and on the training set (0.873, 5-fold *cv*: 0.823 ± 0.04), while F-RSF had lower performance (0.791, 5-fold *cv*: 0.803 ± 0.04). The IBS score was 0.10 for M-RSF and 0.12 for F-RSF.

Figure 1 depicts the plots comparing the KM and predicted survival curves of test subjects (Figure 1a, male MCI patients; and Figure 1b, female MCI patients). M-RSF and F-RSF models showed a large overlap with the KM as demonstrated by low values of RMSE and median and mean absolute error, although a slight decrease in accuracy occurred as time progressed. The bottom plots in Figure 1a,b represent the IBS prediction error per timepoint, where both M-RSF and F-RSF models showed a global maximum at the 24th month but never exceeded the IBS cut-off (dotted red line).

Global explanations on the male MCI training set and the female MCI training set are reported in Figure 2a and Figure 2b, respectively. The rankings of features ordered by their prediction importance are—from the left to the right—RSF feature importance, permutation importance (mean value), and SHAP importance (mean absolute value). Regarding the M-RSF model (Figure 2a), the top three features in the three rankings were FDG, ABETA42, and HCI, while the top three of the F-RSF model were FDG, HCI, and FAQ. Figure 2c depicts the RBO curves of similarity between male and female rankings by increasing depth *d* (RSF M vs. F in plum, Perm M vs. F in violet, SHAP M vs. F in purple). All three pairwise comparisons had low overlap, with a maximum RBO value of 71.1% within the top 12 variables for RSF M vs. F, 60.3% within the top 13 variables for Perm M vs. F, and 67.3% within the top 14 variables for SHAP M vs. F.

Local explanations with SHAP on the test sets are reported in Figure 3. Similarly to global explanations, FDG and HCI were the top features in common between the M-RSF (Figure 3a) and F-RSF models (Figure 3b). The most evident differences in local explanations are the contributions of the hippocampus in M-RSF (+0.09) and RAVLT_perc_forgetting (+0.05), which were absent in F-RSF among the first features. On the contrary, the contribution of LDELTOTAL (+0.09) in the F-RSF model had a low impact on the M-RSF model (+0.03), and the TRABSCOR contribution (+0.08) in the F-RSF model was not among the most contributing variables in the M-RSF model.

The distributions of the risk score in progressing to AD predicted by the M-RSF on test MCI patients are reported as histograms in Figure 4a. Male pMCI test subjects were manually stratified into three risk grades: low range [1.39, 2] (in green), medium range [2, 2.6] (in orange), and high range [2.6, 3.47] (in red). The RSF survival functions of three randomly selected male pMCI subjects per risk grade are shown in Figure 4b. High-risk patient M-pMCI#1 had a risk score of 3.262, converted to AD at the 12th month, and the predicted survival probabilities at each timepoint were [0.89, 0.71, 0.57, 0.43, and 0.28]. Medium-risk patient M-pMCI#2 had a risk score of 1.962, converted to AD at the 24th month, and the predicted survival probabilities at each timepoint were [0.95, 0.83, 0.72, 0.63, and 0.50]. Low-risk patient M-pMCI#3 had a risk score of 1.395, converted to AD at the 36th month, and the predicted survival probabilities at each timepoint were [0.96, 0.87, 0.79, 0.73, and 0.62]. The M-sMCI subject—who does not convert to AD within 36 months—had a risk score of 0.459 and very high predicted survival probabilities per timepoint [0.98, 0.94, 0.92, 0.90, and 0.84].

The distributions of the risk score in progressing to AD predicted by M-RSF on test patients are reported as histograms in Figure 4a. Male pMCI test subjects were manually stratified into three risk grades: low range [1.39, 2] (in green), medium range [2, 2.6] (in orange), and high range [2.6, 3.47] (in red). RSF survival functions of three randomly selected male pMCI subjects per risk grade are in Figure 4b. High-risk patient M-pMCI#1 had a risk score of 3.262, converted to AD at the 12th month, and the predicted survival probabilities at each timepoint were [0.89, 0.71, 0.57, 0.43, and 0.28]. Medium-risk patient M-pMCI#2 had a risk score of 1.962, converted to AD at the 24th month, and the predicted survival probabilities at each timepoint were [0.95, 0.83, 0.72, 0.63, and 0.50]. Low-risk patient M-pMCI#3 had a risk score of 1.395, converted to AD at the 36th month, and the predicted survival probabilities at each timepoint were [0.96, 0.87, 0.79, 0.73, and 0.62]. The M-sMCI subject—who does not convert to AD within 36 months—had a risk score of 0.459 and very high predicted survival probabilities per timepoint [0.98, 0.94, 0.92, 0.90, and 0.84].

In SHAP waterfall plots, a red arrow indicates that the feature increases the risk of conversion from MCI to AD, while a blue arrow indicates that the feature decreases it. The sum of all variable contributions provides the final SHAP value, which corresponds to the prediction risk score. SHAP waterfall plots of M-pMCI#1, M-pMCI#2, M-pMCI#3, and M-sMCI patients are reported in Figure 4c–f, where the actual value of each feature is also reported (in gray). Variables with the highest influence on risk prediction of M-pMCI#1, M-pMCI#2, M-pMCI#3, and M-sMCI subjects were FDG, ABETA42, and HCI (Figure 4c–f), as also found in global and local explanations (Figure 2a and Figure 3a).

Regarding the risk prediction by the F-RSF on female MCI patients, histograms of its distributions are reported in Figure 5a. The stratification per risk grade of female pMCI test was as follows: low range [1.51, 2.3] (in green), medium range [2.3, 3.7] (in orange), and high range [3.7, 5.05] (in red). RSF survival functions of three randomly selected female pMCI subjects per risk grade are depicted in Figure 5b. High-risk patient F-pMCI#1 had a risk score of 4.683, converted to AD at the 6th month, and the predicted survival probabilities at each timepoint were [0.89, 0.62, 0.43, 0.30, and 0.22]. Medium-risk patient F-pMCI#2 had a risk score of 2.799, converted to AD at the 12th month, and the predicted survival probabilities at each timepoint were [0.95, 0.79, 0.67, 0.50, and 0.42].

Low-risk patient F-pMCI#3 had a risk score of 1.51, converted to AD at the 24th month, and the predicted survival probabilities at each timepoint were [0.98, 0.89, 0.84, 0.67, and 0.59]. F-sMCI subject—who does not convert to AD within 36 months—had a risk score of 0.528 and very high predicted survival probabilities per timepoint [0.99, 0.96, 0.94, 0.86, and 0.83].

From Figure 5c–f, which show the SHAP waterfall plots of F-pMCI#1, F-pMCI#2, F-pMCI#3, and F-sMCI patients, it can be noted that variables with the highest influence on risk prediction, such as FDG, HCI, and FAQ, similarly to global and local explanations (Figure 2b and Figure 3b), as well as LDELTOTAL, have a particularly evident contribution in high- and medium-risk patients (Figure 5c,d).

It is worth noting that the average predicted risk by SHAP for male MCI patients (*E*[*f*(*x*)] = 1.769) was lower than the average predicted risk for female MCI patients (*E*[*f*(*x*)] = 2.534).

## 4. Discussion

The present study explored sex-specific differences in predicting the conversion risk from Mild Cognitive Impairment (MCI) to Alzheimer’s Disease (AD) within 3 years using Random Survival Forests (RSF) and SHAP, a model-agnostic approach to boost explainability. The model trained only on male MCI patients (M-RSF) demonstrated optimal performance on both test and training sets, with an accuracy of 0.873 on the test set, while the female model (F-RSF) exhibited slightly lower performance (c-index of 0.791 on the test set), probably due to the lower sample size. Both models displayed low Integrated Brier Scores (IBS), indicating a precise prediction per timepoint. The comparison between Kaplan–Meier and RSF predicted survival curves revealed robust model performance, with high overlap. Despite some common influential features, differences in global and local explanations suggested sex-specific variations in the feature contribution to conversion-to-AD risk prediction. Of note, the average predicted risk for male MCI patients was observed to be lower than that for female MCI patients, underscoring potential sex-specific variations in the risk of conversion from MCI to AD.

Global and local explanations revealed four main common features as influential in both male and female models: FDG, HCI, ABETA42, and FAQ. Very few works have explored the differences between male and female brain glucose metabolism, and findings about gender effect on FDG hypometabolism in normal aging as well as in AD progression are controversial [46]. Overall, a correlation between age and education and brain glucose metabolism was found in temporal and medial frontal regions in healthy adult subjects, but without any significant changes across sexes [46]. This correlation has also been confirmed in AD patients, although males and females exhibited different degrees of association involving different anatomical regions [46]. In our study, feature FDG is calculated as the mean average counting of angular, temporal, and posterior cingulate regions and HCI is a single measurement of FDG-PET hypometabolism (Appendix A), which cannot catch the specific anatomical regional differences between male and female cohorts, as in the past literature [46]. This could explain why FDG and HCI had the same influence in M-RSF and F-RSF models (Figure 2 and Figure 3).

Beta-amyloid (1–42) (feature ABETA42) is a protein that decreases in both the plasma and cerebrospinal fluid (CSF) of dementia patients, which generally does not differ between the sexes [2], in cognitively unimpaired subjects, MCI, or AD patients. Our ML findings confirmed this absence of sex-driven changes in CSF biomarkers, where ABETA42 was among the first four features in the M-RSF and F-RSF, and PTAU and TAU were within the fifteen variables in global and local explanations (Figure 2 and Figure 3).

Feature FAQ provides an assessment of daily living instrumental activities [47] and it is usually administered to the caregiver. In Berezuk et al. [4], FAQ resulted as a significant risk factor for both sexes, although in women it had a stronger effect, which was similarly evident in our results in local explanations (Figure 3). The slightly higher SHAP value of FAQ in the female model (+0.15) than in the male model (+0.08) to the increase in conversion risk score could be associated, as reported by Berezuk et al. [4], with different cognitive reserves across sexes.

In the neural network of learning and memory, the hippocampus plays the role of the central hub, thus its pathological changes contribute to memory impairment [48] resulting in dementia [5]. Differences in hippocampal volume between biological sexes were found in a study that explored amnestic MCI patients, where men had a larger hippocampal volume [49]. Burke et al. investigated the progression from normal cognition to MCI and to probable AD, and they showed that higher hippocampal volumes, or in other words, less hippocampal atrophy, decreased the risk of conversion to AD in women, but it had a more significant role in men [5]. In contrast to the previous studies that highlight hippocampus volume as an important risk factor in the progression from MCI to AD for both sexes [4], we found the hippocampus to have high SHAP values in the male model, while it was completely absent within the first fifteen most important variables in the female model (Figure 2 and Figure 3).

Our two models, M-RSF and F-RSF, also differ in the contribution of two cognitive measures, LDELTOTAL and TRABSCOR, for which SHAP values were higher in the female model than in the male one. LDELTOTAL assesses verbal memory [50], and a plethora of works investigated sex differences in this cognitive domain in normal cognition as well as in dementia [2]. In detail, women score better than men in verbal tasks at all ages, and two studies [10,11] demonstrated that this sex-linked difference persists in the early stages of AD, although other works found that superior female performance in verbal memory tasks is lost in early stages [2]. However, the literature supports that women and men affected by dementia had similar verbal memory scores [2,10,11].

From the local explanations of female MCI patients (Figure 5), it could be noticed how low scores in LDELTOTAL were associated with higher SHAP values in the waterfall plot of high- and medium-risk pMCI patients (Figure 5c,d), while in the low-risk female pMCI and sMCI patient (Figure 5e,f), values of LDELTOTAL higher than 7 decreased the risk to progress to AD within 3 years. Similarly to LDELTOTAL, TRABSCOR, which evaluates executive functions [51], had a higher contribution in the female model than in the male one, where higher scores increased the risk and lower scores decreased it, as depicted in the explanation of female pMCI patients (Figure 5). These results regarding LDELTOTAL and TRABSCOR broadly support the evidence about a stronger decline in memory and executive function in women who progress from MCI to AD compared with men [13].

Differences across sexes in the progression to AD have often been associated with genetic risk factors, like APOE-ε4 allele [7,15]. For example, it has been found that the odds ratio of AD in women with one copy of the APOE-ε4 allele is greater than that in men, as reported in a review about sexual dimorphism in Alzheimer’s disease [15]. However, other studies did not support the role of APOE in differentiating men and women when progressing to dementia [15]. In our study, the feature APOE4 did not have high importance in the male or female model, resulting in a low ranking, as depicted in the global explanations (Figure 2). This is in accordance with the work of Burke et al. [5], who found that APOE-ε4 did not contribute differentially to the progression to MCI or AD among men and women, and with Sohn et al. [3], who found no interaction effects between sex and APOE-ε4.

Summarizing our findings, we find that, similarly to Berezuk et al. [4], the male RSF model had a greater number of CSF and neuroimaging features contributing to the risk prediction, and the female RSF model showed a greater number of neuropsychological tests involved in the progression to AD.

### 4.1. Limitations

We must recognize several limitations in the current work. The study relies on data from the Alzheimer’s Disease Neuroimaging Initiative (ADNI), and as such, the findings may be specific to the characteristics of this particular cohort. Extrapolating the results to broader populations may require caution, as demographic and clinical characteristics can vary across different settings and populations. In detail, we must highlight that the ADNI sample is highly educated, thus we cannot state that in a less educated sample, the same variables would have the same influence. The study considers a fixed timeframe for prediction (36 months). The progression of MCI to AD may exhibit temporal dynamics that extend beyond this timeframe. The study includes a diverse set of biomarkers and features, but the exclusion of certain relevant biological factors or the absence of specific genetic markers might limit the comprehensiveness of the risk prediction models. Another limitation is that this study is not longitudinal, and for this reason, we cannot make causal inferences. Future research endeavors may benefit from collaborative efforts, standardized methodologies, and the integration of diverse datasets to unravel the underlying complexities in this field. Furthermore, our results should be confirmed on independent cohorts of men and women MCI patients to ensure reproducibility.

From a methodological point of view, disparities in the number of stable and progressive MCI patients within the two male and female datasets may have introduced bias. If one class is overrepresented or underrepresented, the model’s performance could be influenced, potentially leading to feature importance rankings that are biased toward the majority class. Efforts to balance the dataset or explore alternative techniques to handle imbalances should be considered, including focusing on the distribution of event occurrences per timepoint. Another methodological issue relies on the fact that the RSF algorithm assumes proportional hazards over time, implying that the hazard ratios remain constant. However, this assumption may not hold in all situations, and violations could impact the accuracy of predictions. A more nuanced examination of time-dependent effects could provide additional insights, for example using model-agnostic methods tailored for survival analysis, such as survLIME [52] or survSHAP [53].

### 4.2. Clinical Implications

Despite the above-described limitations, we believe that our work has important clinical implications. Indeed, we proposed a novel ML approach to investigate sex-specific patterns in AD progression, and as far as we know, it is the first work of this type. Given our optimal performance and robust and stable explanations, we can state that the use of RSF, together with SHAP, represents a valuable tool for personalized interventions and treatment strategies. The different features that SHAP revealed in the study reflect the multifactorial complexity of Alzheimer’s disease, highlighting the importance of considering the interactions between genetic, environmental, and sex-related risk factors.

A detailed understanding of how these features differentially influence disease progression in men and women can aid in identifying individuals at high risk of progression from Mild Cognitive Impairment to AD, significantly enhance therapeutic and diagnostic approaches, and enable the formulation of targeted and personalized preventive interventions.

A key element for both sexes has been identified as the FAQ, which can clinically facilitate the identification of patients requiring support in daily living activities. According to Berezuk et al. [4], women tend to have more experience in these activities, thus developing a greater functional reserve. In line with this theory, one could hypothesize that a personalized medicine approach aimed at enhancing these capabilities and slowing functional decline in both sexes.

For males, important features have emerged as the hippocampus and the RAVLT. The critical role of the hippocampus in memory abilities is well-known, and the RAVLT also assesses this specific cognitive function. Burke et al. [5] found that a reduced hippocampal volume increases the risk of conversion and that stress can accelerate the process of atrophy. Consequently, targeted interventions could be considered to provide tools for managing stressful events in order to prevent atrophy and enhance mnemonic functions as protection against the decline of this cognitive ability.

In women, various cognitive domains such as executive functions, processing speed, and mental flexibility, assessed using the Trail Making Test (TRABSCOR), are relevant. In this case too, it could be useful to enhance these faculties with specific cognitive rehabilitation.

Finally, in addition to genetic factors and biomarkers, various cognitive functions appear to play a significant role; future efforts could aim at implementing targeted and personalized interventions to strengthen those abilities that seem to play a role in the conversion from MCI to AD, in order to prevent or slow cognitive decline.

## 5. Conclusions

In the present work, we applied Random Survival Forests, a Machine Learning technique for survival analysis, to shed light on how and whether feature contributions change according to sex when predicting the risk of progressing to Alzheimer’s disease from Mild Cognitive Impairment. Our results confirmed that women have a higher risk of progressing to dementia; moreover, we highlighted peculiar sex-driven patterns of feature importance with different contributions to the decrease/increase in conversion-to-AD risk. In conclusion, the consistency of these findings with the existing literature underscores established trends, while discrepancies highlight the intricate and multifactorial nature of sex-specific differences in AD progression.

## Figures and Tables

**Figure 1 brainsci-14-00201-f001:**
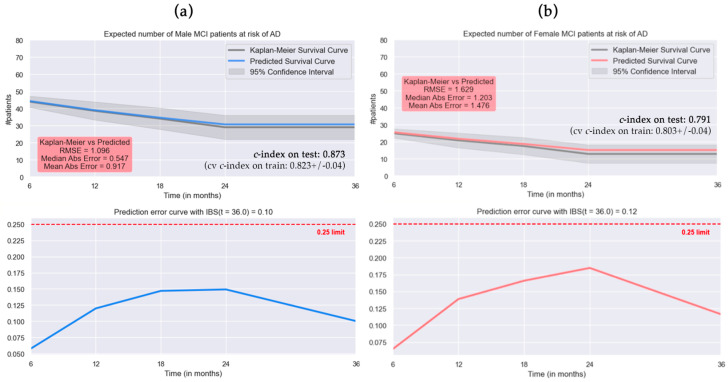
Performance per timepoint on the test set by RSF trained on (**a**) male MCI patients; (**b**) female MCI patients. Upper: plot over time of expected number of MCI patients at risk of conversion to AD, estimated survival curve by Kaplan-Meier in gray. Bottom: Integrated Brier error curve (IBS, critical cut-off limit of 0.25 in red). *C*-index on the test set, cross-validated (*cv*) *c*-index on the training set (mean ± standard deviation), Root Mean Square Error (RMSE), and median and mean absolute error are also reported.

**Figure 2 brainsci-14-00201-f002:**
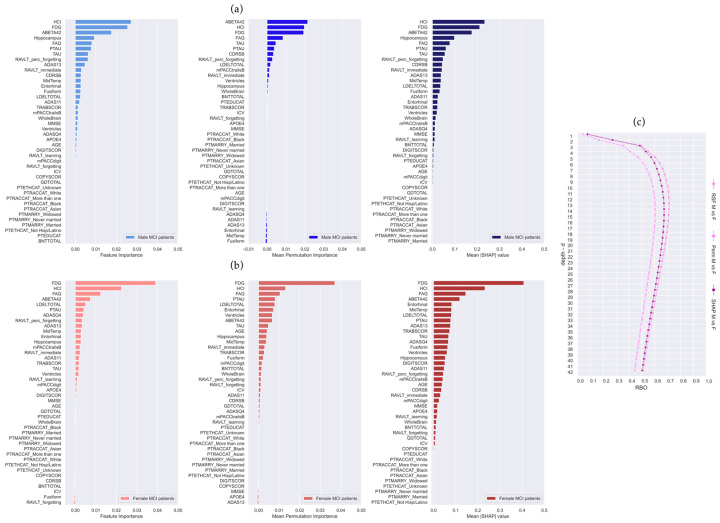
Random Survival Forests (RSF) global explanations trained on (**a**) male MCI (M-RSF) and (**b**) female MCI (F-RSF) patients. From left to right: RSF feature importance (VIMP), permutation importance (mean value), and SHAP importance (mean |SHAP| value). (**c**) Rank-Biased Overlap (RBO) curves assessing the overlap between male and female (M vs. F) variable rankings at different numbers of the important features considered (depth *d*): RSF feature importance (in plum), mean permutation importance (in violet), and mean |SHAP| importance (in purple).

**Figure 3 brainsci-14-00201-f003:**
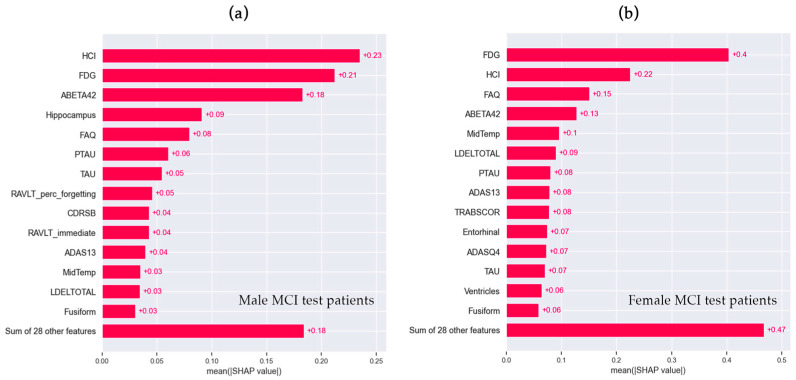
SHAP bar plots of (**a**) male MCI (M-RSF) and (**b**) female MCI test patients (F-RSF).

**Figure 4 brainsci-14-00201-f004:**
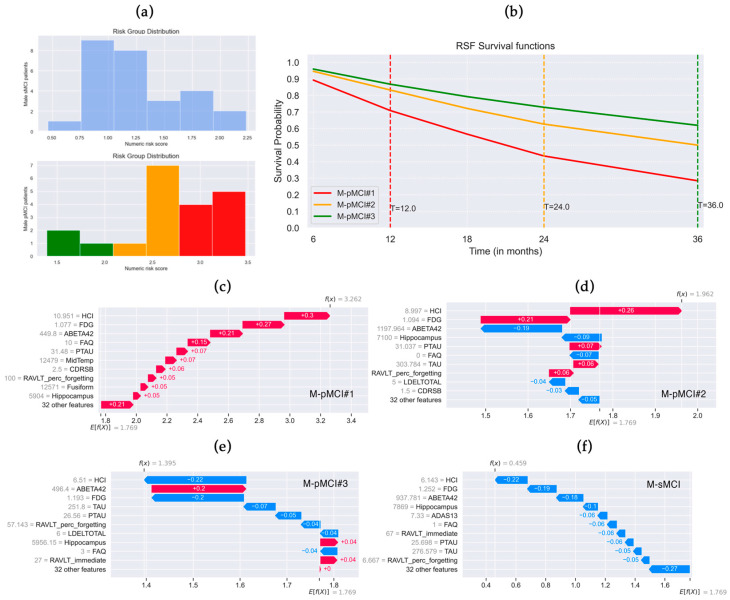
Random Survival Forests (RSF) local explanations trained on male MCI patients. (**a**) Histograms of male sMCI and pMCI patients’ risk distribution predicted by M-RSF. Patients were stratified by risk grade: low (in green, range 1.39–2), medium (in orange, range 2–2.6), high (in red, range 2.6–3.47). (**b**) RSF survival functions of male pMCI patients per risk score: M-pMCI#1 high risk (score 3.262, converted to AD after 12 months), M-pMCI#2 medium risk (score 1.962, converted to AD after 24 months), M-pMCI#3 low risk (score 1.395, converted to AD after 36 months). SHAP waterfall plot of (**c**) patient M-pMCI#1, (**d**) patient M-pMCI#2, (**e**) patient M-pMCI#3, and (**f**) stable MCI patient who does not convert to AD within 36 months (M-sMCI, risk score 0.459). Features that decrease the risk are in blue, while those that increase it are in red. Average predicted risk *E*[*f*(*x*)] = 1.769. Actual value of the feature is in gray.

**Figure 5 brainsci-14-00201-f005:**
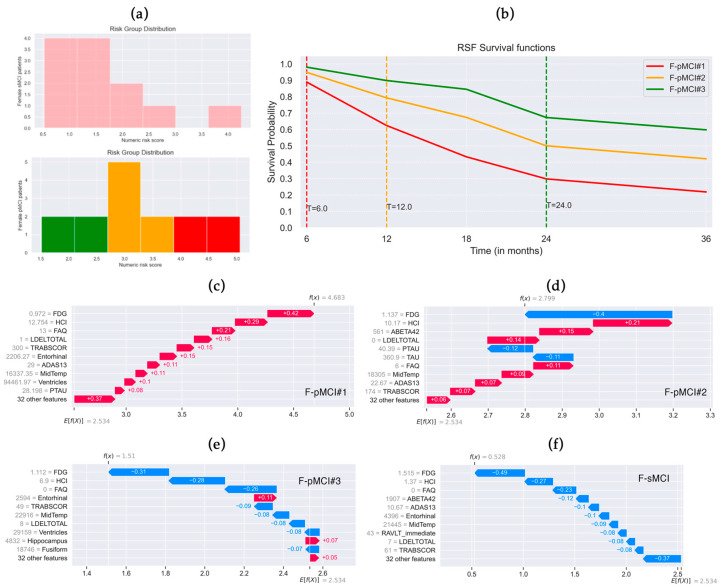
Random Survival Forests (RSF) local explanations trained on female MCI patients. (**a**) Histograms of female sMCI and pMCI patients’ risk distribution predicted by F-RSF. Patients were stratified by risk grade: low (in green, range 1.51–2.3), medium (in orange, range 2.3–3.7), high (in red, range 3.7–5.05). (**b**) RSF survival functions of female pMCI patients per risk score: F-pMCI#1 high risk (score 4.683, converted to AD after 6 months), F-pMCI#2 medium risk (score 2.799, converted to AD after 12 months), F-pMCI#3 low risk (score 1.51, converted to AD after 24 months). SHAP waterfall plot of (**c**) patient F-pMCI#1, (**d**) patient F-pMCI#2, (**e**) patient F-pMCI#3, and (**f**) stable MCI patient who does not convert to AD within 36 months (F-sMCI, risk score 0.528). Features that decrease the risk are in blue, while those that increase it are in red. Average predicted risk *E*[*f*(*x*)] = 2.534. Actual value of the feature is in gray.

**Table 1 brainsci-14-00201-t001:** Demographic, clinical, cognitive, CSF, and imaging data of sMCI and pMCI patients stratified by sex.

	M (233)		F (132)	
	sMCI (136)	pMCI (97)	sMCI (62)	pMCI (70)
**Demographic:**				
*Age*	74.7 ± 7.3	74.6 ± 7.5	75.8 ± 7.6	74.6 ± 6.0
*Education level*	15 ± 3.3	15.4 ± 2.9	16.2 ± 2.6	16.1 ± 2.8
**Biomarker:**				
*APOE4 (0/1/2)*	82/42/12	38/43/16	28/26/8	16/43/11
**Clinical scale:**				
*CDRSB*	1.4 ± 0.8	1.9 ± 0.9	1.6 ± 0.9	1.8 ± 1.1
*FAQ*	2.3 ± 3.2	5.6 ± 5.3	2.8 ± 4.1	5.6 ± 4.5
**Neuropsychological assessment:**				
*ADAS11*	10.5 ± 4.2	13.12 ± 3.8	10.5 ± 4.6	13.2 ± 4.5
*ADAS13*	16.8 ± 6.0	21.3 ± 5.0	16.9 ± 6.7	21.1 ± 6.2
*ADASQ4*	5.6 ± 2.2	7.13 ± 1.9	5.5 ± 2.4	7.1 ± 2.0
*MMSE*	27.3 ± 1.8	26.68 ± 1.7	27.2 ± 1.7	26.6 ± 1.8
*RAVLT_immediate*	33.1 ± 9.6	27.25 ± 6.9	33.1 ± 10.6	27.2 ± 6.2
*RAVLT_learning*	3.8 ± 2.3	2.74 ± 1.9	3.7 ± 2.3	3.0 ± 2.0
*RAVLT_forgetting*	4.5 ± 2.4	4.86 ± 2.09	4.5 ± 2.3	5.0 ± 2.2
*RAVLT_perc_forgetting*	59.9 ± 31.5	77.62 ± 27.6	63.8 ± 31.0	79.0 ± 28.5
*LDELTOTAL*	4.3 ± 2.7	2.8 ± 2.4	4.8 ± 2.5	3.3 ± 3.1
*DIGITSCOR*	38.9 ± 11.1	34.13 ± 11.2	37.0 ± 9.7	34.0 ± 10.5
*TRABSCOR*	116.4 ± 64.2	146.85 ± 79.9	125.6 ± 67.1	149.9 ± 80.2
*mPACCdigit*	−3.9 ± 3.9	−3.9 ± 4.8	−3.8 ± 3.8	−3.9 ± 4.9
*mPACCtrailsB*	−3.8 ± 4.0	−3.7 ± 4.8	−3.8 ± 3.8	−3.0 ± 4.8
*GDTOTAL*	1.6 ± 1.4	1.54 ± 1.3	1.6 ± 1.3	1.6 ± 1.4
*COPYSCOR*	4.7 ± 0.7	4.5 ± 1.2	4.6 ± 0.8	4.7 ± 0.6
*BNTTOTAL*	25.5 ± 4.0	24.55 ± 4.6	26.2 ± 3.2	25.7 ± 3.6
**CSF:**				
*ABETA42*	1027.5 ± 398.4	676.7 ± 224.3	881.4 ± 364.2	708.8 ± 309.2
*TAU*	307.2 ± 89.4	316.43 ± 73.2	307.5 ± 110.9	331.0 ± 90.2
*PTAU*	30.4 ± 10.8	31.76 ± 8.0	31.2 ± 14.0	33.3 ± 10.8
**Neuroimaging:**				
*Ventricles*	41,196.5 ± 24,245.5	44,937.2 ± 18,888.8	48,192.5 ± 26,633.5	50,164.0 ± 27,112.5
*Hippocampus*	6699.8 ± 987.2	5862.6 ± 923.9	6452.7 ± 963.8	6092 ± 1095.6
*WholeBrain*	1,005,263.4 ± 106,966.9	973,259.03 ± 115,953.1	1,013,898.5 ± 100,967.5	990,467.4 ± 111,353.5
*Entorhinal*	3480.9 ± 711.5	2997.0 ± 698.9	3475.5 ± 707.2	3031.5 ± 723.4
*Fusiform*	16,831.29 ± 2179.3	15,618.0 ± 2409.4	16,947.7 ± 2161.6	15,884.9 ± 2393.1
*MidTemp*	19,134.46 ± 2554.2	17,311.45 ± 3127.2	19,604.9 ± 2652.1	17,911.5 ± 2708.5
*ICV*	1,562,645.6 ± 163,009.0	1,554,468.3 ± 170,366.0	1,609,215.3 ± 162,431.2	1,587,939.1 ± 176,315.7
*FDG*	1.2 ± 0.1	1.07 ± 0.1	1.2 ± 0.1	1.1 ± 0.1
*HCI*	7.1 ± 2.8	9.54 ± 2.5	7.1 ± 2.7	9.9 ± 2.9
**Occurrence of event (pMCI = 1) and censorship (sMCI = 0) per timepoint** (in months):				
*m06*	17	14	3	8
*m12*	12	27	5	20
*m18*	14	20	7	15
*m24*	21	18	10	18
*m36*	72	18	37	8

Mean and standard deviation are calculated after imputation of missing data. For abbreviations see Appendix A.

**Table 2 brainsci-14-00201-t002:** Statistical analysis between male and female MCI patients.

		*p*-Value		
	M-sMCI vs. M-pMCI	F-sMCI vs. F-pMCI	M-sMCI vs. F-sMCI	M-pMCI vs. F-pMCI
**Demographic:**				
*Age*	0.95 ^a^	0.32 ^a^	0.35 ^a^	0.96 ^a^
*Education level*	0.27 ^a^	0.81 ^a^	**0.006** ^a^	0.14 ^a^
**Biomarker:**				
*APOE4 (0/1/2)*	**0.005** ^b^	**0.02** ^b^	0.14 ^b^	0.58 ^b^
**Clinical scale:**				
*CDRSB*	**<0.001** ^c^	0.12 ^c^	0.11 ^c^	0.90 ^c^
*FAQ*	**<0.001** ^c^	**<0.001** ^c^	0.34 ^c^	0.93 ^c^
**Neuropsychological assessment:**				
*ADAS11*	**<0.001** ^c^	**<0.001** ^c^	0.95 ^c^	0.93 ^c^
*ADAS13*	**<0.001** ^c^	**<0.001** ^c^	0.97 ^c^	0.78 ^c^
*ADASQ4*	**<0.001** ^c^	**<0.001** ^c^	0.67 ^c^	0.94 ^c^
*MMSE*	**0.01** ^c^	**0.03** ^c^	0.82 ^c^	0.76 ^c^
*RAVLT_immediate*	**<0.001** ^c^	**<0.001** ^c^	0.87 ^c^	0.92 ^c^
*RAVLT_learning*	**<0.001** ^c^	**0.03** ^c^	0.88 ^c^	0.46 ^c^
*RAVLT_forgetting*	0.24 ^c^	0.3 ^c^	0.82 ^c^	0.64 ^c^
*RAVLT_perc_forgetting*	**<0.001** ^c^	**0.003** ^c^	0.41 ^c^	0.76 ^c^
*LDELTOTAL*	**<0.001** ^c^	**0.004** ^c^	0.28 ^c^	0.28 ^c^
*DIGITSCOR*	**0.001** ^c^	0.06 ^c^	0.31 ^c^	0.93 ^c^
*TRABSCOR*	**0.001** ^c^	0.06 ^c^	0.43 ^c^	0.81 ^c^
*mPACCdigit*	0.94 ^c^	0.97 ^c^	0.86 ^c^	0.98 ^c^
*mPACCtrailsB*	0.89 ^c^	0.21 ^c^	0.89 ^c^	0.35 ^c^
*GDTOTAL*	0.77 ^c^	0.98 ^c^	0.85 ^c^	0.87 ^c^
*COPYSCOR*	**0.03** ^c^	0.76 ^c^	0.72 ^c^	0.07 ^c^
*BNTTOTAL*	0.09 ^c^	0.33 ^c^	0.22 ^c^	0.07 ^c^
** *CSF:* **				
*ABETA42*	**<0.001** ^c^	**0.003** ^c^	**0.015** ^c^	0.43 ^c^
*TAU*	0.41 ^c^	0.17 ^c^	0.97 ^c^	0.25 ^c^
*PTAU*	0.29 ^c^	0.32 ^c^	0.67 ^c^	0.29 ^c^
**Neuroimaging:**				
*Ventricles*	0.067 ^d^	0.11 ^d^	0.50 ^d^	0.33 ^d^
*Hippocampus*	**<0.001** ^d^	**0.01** ^d^	0.053 ^d^	0.27 ^d^
*WholeBrain*	0.02 ^c^	0.10 ^c^	0.38 ^c^	0.32 ^c^
*Entorhinal*	**<0.001** ^d^	**<0.001** ^d^	0.75 ^d^	0.91 ^d^
*Fusiform*	**<0.001** ^d^	**0.005**	0.71 ^d^	0.96 ^d^
*MidTemp*	**<0.001** ^d^	**0.001** ^d^	0.54 ^d^	0.47 ^d^
*ICV*	0.70 ^c^	0.45 ^c^	0.063 ^c^	0.22 ^c^
*FDG*	**<0.001** ^c^	**<0.001** ^c^	0.66 ^c^	0.94 ^c^
*HCI*	**<0.001** ^c^	**<0.001** ^c^	0.96 ^c^	0.34 ^c^

In bold: significant result at *p* < 0.05. ^a^ One-way ANOVA; ^b^ Chi-square test; ^c^ ANCOVA with age in covariates; ^d^ ANCOVA with age and ICV in covariates. For abbreviations, see Appendix A.

**Table 3 brainsci-14-00201-t003:** Hyperparameters of Random Survival Forests (RSF) trained on male MCI (M-RSF) and female MCI (F-RSF) patients. Optimal values of hyperparameters were obtained through a randomized search with 3-fold cross-validation and 50 repetitions.

		Optimal Value	
Hyperparameter	Parameter Distribution	M-RSF	F-RSF
*max_depth*	integer from a *reciprocal continuous random distribution* in range (5, 50)	26	42
*min_node_size*	integer from a *reciprocal continuous random distribution* in range (1, 40)	34	19
*max_features*	[‘all’, ‘sqrt’, ‘log2’]	‘sqrt’	‘sqrt’
*sample_size_pct*	[0.60, 0.70, 0.80, 0.90]	0.70	0.60

## Data Availability

Data used in the preparation of this article were obtained from the Alzheimer’s Disease Neuroimaging Initiative (ADNI) database (adni.loni.usc.edu). Accessed on 5 June 2023.

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
