# Peer review of "Sex Differences in Conversion Risk from Mild Cognitive Impairment to Alzheimer’s Disease: An Explainable Machine Learning Study with Random Survival Forests and SHAP"

_brainsci, 2024, doi:10.3390/brainsci14030201_

Round 1
Reviewer 1 Report
Comments and Suggestions for Authors
This study investigates difference between women and men regarding conversion risk from mild mental to Alzheimer.
This study is timely with high significance.
Though, the report shows some weaknesses.
1. Abstract
I miss the motivation and objectives of the study
2. Figures
See attached document
Figures are not readable.
3. Machine learning method
Explain why you choose the rsf as method.
The paper needs minor revision

Author Response
We want to thank both Reviewers for their precious comments and suggestions. We believe that following their valuable opinion our manuscript has been improved. We have replied point by point, and changes in the manuscript have been highlighted in yellow. Due to the long Discussion, we added a section Conclusions. Please, note that several parts of the main text have been modified to avoid high similarity with our recent published work by Sarica et al. 2023. Attached the reply in pdf format.

Reviewer 2 Report
Comments and Suggestions for Authors
The manuscript explores the gender-specific risk factors influencing the progression from Mild Cognitive Impairment (MCI) to Alzheimer's Disease (AD), utilizing Explainable Machine Learning techniques with Random Survival Forests and SHAP. This approach aims to shed light on intricate mechanisms underlying AD progression, which could be pivotal for developing gender-specific interventions.
Major Comments:
1- The manuscript's methodology section lacks detailed explanation on the feature selection process and the criteria for model validation. Given the complexity of AD progression, it's crucial to understand how the model accounts for variables beyond gender differences, such as genetic predispositions and environmental factors.
2- While the application of SHAP values offers insight into model interpretability, there's a need for a more comprehensive discussion on the clinical relevance of the identified predictive features. Specifically, how do these features align or conflict with current understanding in neurology and geriatrics literature?
3- The analysis seems to overlook potential confounders that could influence the progression from MCI to AD. A deeper examination of how these confounders were controlled or their impact assessed would strengthen the manuscript's validity.
4- Could you elaborate on how the dataset was compiled, specifically regarding the inclusion and exclusion criteria for subjects’ data? Additionally, how do you address potential biases arising from the dataset’s composition?
Minor Comments:
1- Figures and tables presented to support the findings could benefit from clearer descriptions and annotations to enhance their interpretability for readers not familiar with machine learning.
2- The manuscript would be improved by a section discussing the limitations of the study, particularly in terms of the model's generalizability to diverse populations outside the study's dataset.
The manuscript contributes valuable insights into the gender-specific factors influencing AD progression, leveraging cutting-edge machine learning techniques. However, to solidify its findings and enhance its impact, addressing the major concerns raised regarding methodology, data analysis, and interpretability is recommended. Incorporating a more detailed discussion on the clinical implications of the predictive features identified could also provide a significant contribution to personalized medicine in AD research.
Comments on the Quality of English Language
The manuscript is well-written, with a clear and coherent structure that facilitates understanding. However, attention to minor grammatical errors and consistency in terminology will further enhance its readability and professionalism.
Author Response

(The authors gave the same response as above.)
